#### Metal Layer Depletion during the Super Substorm on 4 1

#### **November 2021** 2

- Gang Chen<sup>1\*</sup>, Yimeng Xu<sup>1</sup>, Guotao Yang<sup>2</sup>, Shaodong Zhang<sup>1,3</sup>, Zhipeng Ren<sup>4</sup>, Pengfei 3
- Hu<sup>1</sup>, Tingting Yu<sup>4</sup>, Fuju Wu<sup>5</sup>, Lifang Du<sup>2</sup>, Haoran Zheng<sup>2</sup>, Xuewu Cheng<sup>6</sup>, Faquan Li<sup>6</sup>, 4
- Min Zhang<sup>1</sup> 5
- <sup>1</sup>School of Earth and Space Science and Technology, Wuhan University, Wuhan 430072, China
- <sup>2</sup>National Space Science Center, Chinese Academy of Sciences, Beijing 100190, China
- <sup>3</sup>School of Physics and Electronic Science, Guizhou Normal University, Guiyang 550025, China
- <sup>4</sup>Key Laboratory of Earth and Planetary Physics, Institute of Geology and Geophysics, Chinese
- Academy of Sciences, Beijing 100029, China
- <sup>5</sup>School of Physics, Henan Normal University, Xinxiang 453007, China
- <sup>6</sup>Innovation Academy for Precision Measurement Science and Technology, Chinese Academy of
- Sciences, Wuhan 430071, China

21

24

- 14 Corresponding Author: Gang Chen (g.chen@whu.edu.cn)
- 15 Abstract. Metal layer forms as a result of meteoric ablation and exist as a layer of metal elements 16 between approximately 80 and 105 km altitude, and it provides information about the physics and 17 chemistry of the boundary between the atmosphere and space. There are some studies about the wind 18 field disturbances in Mesosphere and Low Thermosphere (MLT) region and the plasma variations in 19 ionospheric E-region during magnetic storms, but no study on the impact of storms on the metal atom 20 layers in mesosphere. During the super substorm on 4 Nov. 2021, the atmospheric metal layers were observed to decrease by observations from three lidars at the mid-latitudes of China. The Na, Ca and Ni 22 densities on the storm day were significantly lower than in other days in October and November. The 23 O/N<sub>2</sub> column density ratio observed by the Global Ultraviolet Imager (GUVI) on the storm day was much higher than on quiet days, and the numerical simulation results demonstrate a substantial increase 25 in atomic oxygen density at the heights of the metal layer. The increase in oxygen density may lead to 26 the formation of more metal compounds, thus more metal atoms are consumed. This is an interesting phenomenon that magnetic storm can perturb the atmospheric metal layer through chemical reactions.

#### 1 Introduction

Earth's metal layer is located at the height of the Kármán line, namely the boundary separating Earth's atmosphere and outer space. It is composed by the metallic elements, such as Fe, Mg, Na, K, Ca, Ni, etc, which are due to meteoroid ablation through collision with atmospheric molecules (Plane et al., 2015). These metallic elements undergo complex chemical reactions with atmospheric composition as well as dynamic processes with background wind field, eventually they are highly concentrated between 80-105 km altitude and form a metal layer that envelops the earth (Gardner et al., 2005, 2011; Hunten, 1967; Kopp, 1997). Components of the metal layer turn into compounds that deposit on the Earth's surface in about four years, and become the nourishment of the biosphere (Plane et al., 2015). Our knowledge of metal layers is still very limited, especially the mechanism for its distribution features in the Earth's atmosphere. Temporal variation of the metal layer is of primary concern, therein, Na layer is the most studied due to its high concentration of atomic particles and large scattering cross section. The density of Na peaks in winter, and reaches its lowest level in summer at all latitudes (Gong et al., 2013; Yi et al., 2009). It is widely accepted that the Na layer exists at an altitude of 80-105 kilometers, and at lower altitudes, Na atoms are mostly converted into compounds due to chemical reactions. The background temperature and wind field at these heights are suggested to be responsible for the Na layer variation (Hickey and Plane, 1995). The density variation of Ca layer has two peaks in winter and summer, and the density peak in winter is higher (Gerding et al., 2000; Xun et al., 2024). It suggests to be relevant to the meteor input function (Plane et al., 2018). The density of Ni layer is maximum in winter and minimum in summer, similar to the seasonal cycle of Na layer (Jiao et al., 2022). At middle and high latitudes, the density of Fe layer peaks in winter and is lowest in summer. Conversely, at low latitudes, Fe layer density reaches its maximum in spring and becomes lowest in the fall. The global distribution features of the mesospheric ozone and O<sub>2</sub> is attributed to result in these latitudinal differences (Yi et al., 2009; Raizada et al., 2003). K layer is different to other layers, its density is maximum in winter and summer, and minimum in spring and autumn (Eska et al., 1998; Yue et al., 2017). There is a correlation between the Na and Ni layers, but not with other metal layers (Wu et al., 2022). Solar flare and coronal mass ejection can induce geomagnetic storm on the earth, including intense currents in magnetosphere, changes in radiation belts, variations in the ionosphere, and heating

of the thermosphere. Although the wind field and electric field disturbances in storms are hard to penetrate deep into the dense atmospheric region due to the viscous force of air, the wind field disturbances in Mesosphere and Low Thermosphere (MLT) region as well as the plasma variations in ionospheric E-region have been recorded and analyzed (Chen et al., 2023; Li et al., 2019; Li et al., 2024; Resende et al., 2020). The metal layer is also located in MLT region, and the variations of the metal atom density in the metal layer can be observed by lidar in the night with good weather condition. During the intense magnetic storm on 4 Nov. 2021, we have recorded the density decrease of different compositions in the metal layers at two locations, and this is the first time the influence of a storm on the metal atoms in the MLT region is recorded.

#### 2 Data and Analysis Methods

In Oct. and Nov. 2021, two lidars in Yangqing, Beijing, China (40.42°N, 116.02°E) conducted the observations of Na, Ni and Ca layers, and one lidar in Pingquan, Hebei, China (41°N, 118.7°E) conducted the observations of Na layer. The Pingquan station is 237 km east of the Yanqing station. On the storm day of 4 Nov. 2021, all the three lidars were well operated. The Na and Ni observations were implemented by the dual-wavelength simultaneous detection system (Du et al., 2020; Jiao et al., 2015; Wu et al., 2021). The all-solid-state narrowband lidar is used to observe the Ca layer (Du et al., 2023). The time and height resolutions of these lidars are 33s and 96m, respectively. After long time integration to improve the Signal-to-Noise Ratio (SNR), the time resolutions of the output data are one hour. Due to the very low received signal power, the range integration is executed for the Ni layer observation and the range resolution of the output data becomes one kilometer.

The detection threshold of the lidars is defined as the background noise plus two times the noise standard deviation (Gao et al., 2015). The error of the lidar photon counting is inversely proportional to the square root of the return photon number, and the error of the measured metal atom density determined by the error of the photon counting is no more than 5%.

The absolute density can be estimated by the following standard lidar equation (Megie et al., 1978;

Gerding et al., 2018; Chu and Papen, 2005),

$$83 n_{atom}(z) = n_R(z_R) \frac{N_s(\lambda, z) - N_B \Delta t}{N_R(\lambda, z_R)} \frac{z^2}{-N_B \Delta t} \frac{\sigma_R(\lambda)}{z_R^2} \frac{\sigma_R(\lambda)}{\sigma_{atom}(\lambda)}$$
 (F1)

The reference altitude  $z_R$  is usually selected between 30-45 km, and z represents the altitude of the

resonant fluorescence scattering region, and  $n_R(z_R)$  is the atmospheric total number density at the reference altitude, which can be obtained from the NRLMSISE-00 model.  $N_s(\lambda,z)$  and  $N_R(\lambda,z_R)$  are the return photon numbers from the altitude z and the reference altitude  $z_R$ , respectively.  $N_B$  is the return photon number from the noise altitude generally at 150-190 km.  $\lambda$  represents the wavelength of  $N_B$  (589.158 nm),  $N_B$  (2422.6728 nm), and  $N_B$  (341.5744 nm). At 30 km altitude, the Rayleigh backscattering cross section  $\sigma_R(589.158nm)$  for  $N_B$  is  $4.14\times10^{-32}$  m<sup>2</sup>sr<sup>-1</sup> and the effective backscattering cross section of resonance  $\sigma_{atom}(589.158nm)$  for  $N_B$  is  $5.22\times10^{-16}$  m<sup>2</sup>sr<sup>-1</sup>, the  $\sigma_R(422.6728nm)$  for  $N_B$  is  $1.28\times10^{-31}$  is m<sup>2</sup>sr<sup>-1</sup> and the  $\sigma_{atom}(422.6728nm)$  for  $N_B$  is  $1.58\times10^{-17}$  m<sup>2</sup>sr<sup>-1</sup>. When calculating the absolute density of the  $N_B$  layer, it is also necessary to consider the branching fraction of the transition lines of the received return photons (Wu et al., 2021).

The number of return photons should be proportional to the flux of the transmitted laser photons. However, for large laser intensities, the returned fluorescence photons are no longer proportional to the laser power, but less than expectations. Thus, the saturation effect occurs. The content of Ni atoms in metal layer is relatively low, so its saturation effect is generally not considered. The energy of Na lidar's laser is not high enough to produce saturation effect. The laser energy emitted by the Ca lidar is relatively large and the divergence angle of the laser beam is very small, thus the saturation effect of the Ca lidar must be considered. Refer to the method of Megie et al., (1978), Welsh and Gardner (1989), and Chu and Papen (2005) to estimate saturations, the ratio  $\frac{N^{\text{Sat}}(z)}{N(z)} = 91.42\%$  is obtained (Wu et al., 2020), where  $N^{\text{Sat}}(z)$  is the number of the photons received at height z, when the saturation effect occurs, and N(z) is the number of photons received at height z, when there is no saturation effect. Thus, we have taken this ratio into account in the effective scattering cross sections to correct the Ca densities.

All the measured metal densities, including the density values of different metal atoms at different places, are quantitative and can be compared directly.

GUVI is a spectrograph imager on the Thermosphere Ionosphere Mesosphere Energy and Dynamics (TIMED) spacecraft. It can provide cross-track scanned images of the Earth's airglow and auroral emission in the Far UltraViolet (FUV) at wavelengths between 110.0 and 185.0 nm (Christensen et al., 2003). The OI 135.6 nm and N<sub>2</sub> Lyman-Birge-Hopfield (140-150 nm) data were

extracted from the FUV spectral data and can be used to calculate the global thermospheric column density  $O/N_2$  ratio,  $\Sigma O/N_2$  (Zhang et al., 2004). The global  $O/N_2$  variations can be used to describe the response of thermosphere-neutral components to geomagnetic activities.

Thermosphere Ionosphere Electrodynamics General Circulation Model (TIEGCM) is a self-consistent general circulation model of the coupled ionosphere-thermosphere system. It can solve the three-dimensional energy, continuity, momentum and electrodynamic equations of ions and neutral species (Richmond et al., 1992; Roble et al., 1988). The altitude range of the model is between 97 and 600 km and the spatial resolution is  $1.25^{\circ} \times 1.25^{\circ} \times 0.25$  scale heights. The bottom boundary of TIEGCM is ~97 km, solving a series of equations in the thermosphere and ionosphere system self-consistently. Solar extreme ultraviolet irradiance and the auroral electron precipitation are input in this model (Solomon and Qian, 2005; Roble and Ridley, 1987). The Heelis empirical model (Heelis et al., 1982) and the monthly tidal climatology (Hagan and Forbes, 2002, 2003) are used as the high-latitude and the lower boundary inputs, respectively.

### **3 Results**

## 3.1 Geomagnetic index

Figure 1. Geomagnetic indexes. (a) Bz, (b) SYM/H, (c) AE, (d) PC and (e) kp indexes on 3-5 November 2021.

The green and red dashed boxes indicate the observation periods of the lidars in the nights of 3 and 4 Nov..

The red up arrow in the bottom plot is used to indicate the beginning of the simulated O density

enhancement in MLT region.

Figure 1 shows the variations of Bz, SYM-H (can be seen as a high-resolution Dst), AE, PC and Kp indices on the three days of 3-5 Nov. 2021 from top to bottom. These geomagnetic indexes can be freely accessed in the OMNIWeb (1963). Before 17:00 UT on 3 Nov., there was no geomagnetic activity. The Interplanetary Magnetic Field (IMF) Bz component in Fig. 1a began to turn southward

firstly at ~20:00 UT indicating the beginning of the storm and then it turned southward and northward for many times until ~12:38 UT on the next day. The SYM-H index in Fig. 1b shows that the Storm Sudden Commencement (SSC) began at ~20:00 UT on 3 Nov. and the storm continued to 5 Nov. The AE index in Fig. 1c presented several peaks. The first peak emerged at 21:45 UT on 3 Nov. with the maximum value of 2379 nT, and many AE peaks followed. There are two peaks exceeded 3000 nT at 9:13 UT and 11:25 UT on 4 Nov., indicating the occurrence of super substorm. The variation of the Polar Cap (PC) index of the Northern Hemisphere in Fig. 1d can be used as an indicator of the level of Joule heating. The PC index has exceeded 10 mV/m for many times, and the maximum PC index of 18.43 mV/m emerged at ~22:00 UT on 3 Nov., indicating the polar region Joule heating events have occurred many times, including one intense heating event at least. The Kp index used to characterize the magnitude of geomagnetic storms is shown in Fig. 1e. The Kp index exceeded 5 at 21:00 UT on 3 Nov. and reached its maximum of 7.7 at 9:00 UT on 4 Nov., indicating the strong geomagnetic storm happened. The green and red dashed boxes in Figure 1 are used to indicate the observation periods of the lidars in the nights of 3 and 4 Nov.. The geomagnetic indexes show that the influence of the storm began in the daytime of 4 Nov., at that time, the lidars didn't work.

# 3.2 Observation results of the lidars

Figure 2. Lidar observations of the metal layers. The lidar observations of the Na density in Pingquan, as well as the Na, Ca and Ni density in Yanqing are listed from top to bottom row. The observations on the substorm day of 4 Nov. 2021 are displayed in the third column and highlighted by the red dashed box. The observations on the reference days are shown in the second and forth columns. The monthly average of November is shown in the fifth column. The first column shows the altitude profiles of the average metal density for a day, and the error bar indicates the uncertainty in density measurements. LT = UT + 8 h.

Morphological structures of the metal layers between 75-120 km height in the night are displayed in Fig. 2. The metal atoms dataset is accessed at the Zenodo website (Xu and Chen, 2025). The observations on the storm day of 4 Nov. 2021 are shown in the third column as highlighted by the red dashed box. The observations on the reference days are shown in the first and third columns. The monthly average of the observations in Nov. 2021 is displayed in the fifth column. The observed Na layer in Pingquan, the Na, Ca, and Ni layers in Yanqing are shown from top to bottom rows in sequence. The observation time of the lidars is set between 10 and 21 UT (18 and 5 LT of the next day), usually, and the weather conditions will determine the exact time of observation. Compared to the conference days and the monthly average, the density of the Na layer density in Pingquan, the Na, Ca and Ni layer density in Yanging were obviously lower on 4 Nov. The maximum Na density in Figs. 2b and 2f was no more than 3600 atoms cm<sup>-3</sup>, and the Na layers were distributed in a narrow altitude range of 90-96 km, but the Na layer on other days emerged in wider altitude range with much higher peak metal density. The Ca layer in Fig. 2j on 4 Nov. almost disappeared and the maximum Ca density was as lower as 16.6 atoms cm<sup>-3</sup>. The Ca layer presented very high density on the reference days, and the peak density had exceeded 80 atoms cm<sup>-3</sup>. For a long time in Fig. 2i, the monthly average of Ca layer density exceeded 40 atoms cm<sup>-3</sup>. The maximum density of the Ni layer in Fig. 2n was lower than 110 atoms cm<sup>-3</sup>, also lower than on the reference days in Figs. 2m and 2o, and much lower than monthly averages in Fig. 2p.

The average altitude profiles of the measured metal layer density in a day are displayed in the first column of Figure 2. The average profiles recorded on the storm day are shown as red curves, those recorded on the reference days before and after are shown as blue and green dashed curves, respectively, and the average profiles of the monthly average are shown as the gray dashed curves. There is an error in the measurement of photon counts by a lidar, thus there is also uncertainty in metal density

measurements. The error bars on the profiles are used to indicate the uncertainty of the estimated metal density at different altitudes. The maximum Na, Ca and Ni densities on the storm day in Figure 2q, 2s 2t are lower than on reference days and month averages, and the their height distributions are also narrower. The maximum density of the red Na density profile in Figure 2r is lower than the green and gray profiles, but its is almost equal to that of the blue profile. The height distribution of red profile is narrower than the blue one, thus the Na layer density on the storm day was generally lower than on 3 Nov.

Figure 3. Average column abundances of the metal layers. The observations of the (a) Na lidar in Pingquan, as well as the (b) Na lidar, (c) Ca lidar and (d) Ni lidar in Yanqing are listed from top to bottom row. The red bars indicate the observations on the substorm day. Each vertical bar shows the average column abundance between 75-120 km altitude in one night and all the valid observation results in October and November 2021

### are displayed. The uncertainty of the column abundance is exhibited on the top of each bar.

Figure 3 presents the average column abundance of the metal layers on each observation day of Oct. and Nov. 2021 and the observations of the Na column abundance in Pingquan, the Na, Ca and Ni column abundance in Yanqing are listed from top to bottom in sequence. The average column abundance of a day can be obtained by integrating the density profile in the first column of Figure 2 along the height between 75-120 km. The error bar on the top of each bar shows the uncertainty of the column abundance. The uncertainty is about one thousandth of the column abundance, thus error bar is compressed into almost a horizontal line.

In Oct. and Nov. 2021, except for the storm on 4 Nov., there was only a moderate storm occurring on 12 Oct. with the Kp index of 6.3 and all the lidars were not operated on that day. There is no obvious regularity in the day-to-day variation of the metal layer in the two months, but the lowest column abundances in all four plots of Fig. 3 occurred on 4 Nov. 2021 as shown by the red bars. We have estimated the correlations between the different metal layer data in Fig. 3, and the correlation coefficients of Pingquan Na-Yanqing Na, Pingquan Na-Yanqing Ca, Pingquan Na-Yanqing Ni, Yangqing Na-Ca, Yangqing Na-Ni, and Yangqing Ca-Ni are 0.397, 3.578×10<sup>-4</sup>, 0.507, 0.346, 0.603 and 0.169, respectively. There is little correlation between them. However, this does not mean that they are not correlated over a longer time scale. If there is no magnetic storm, the probability of all the four observations recording the lowest column abundances on the same day is 1.334×10<sup>-7</sup>. Therefore, the metal layer depletion on 4 Nov. 2021 should be related to the storm.

We also can find that the abundance variation of the Ca has presented less correlation with other atoms, and the correlation coefficient of Pingquan Na-Yanqing Ca is much lower. Observations indicate that the Ni and Na layers show close correlations on the scale of hours (Wu et al., 2022). And the calcium layer is somewhat unique. Although Ca abundance has a similar elemental abundance to Na in meteorites, the Ca atom abundance in metal layer is roughly 2 orders of magnitude smaller than Na. Plane et al., (2018) suggested that CaOH and CaCO3 are stable reservoirs for Ca in metal layer, as a result, more Ca atoms are converted to compounds and the Ca atom abundance in metal layer is much less than the Na atom abundance. The Ca abundance variation is more affected by chemical reactions, so it has less correlation with other metal atoms. The relevance between the Ca and Na in different places further decreases, thus the estimated correlation coefficient becomes very low.

## 3.3 Observation results of the GUVI

Figure 4. GUVI O/N<sub>2</sub> distribution in the Eastern Hemisphere. The observations on (a) 3 November, (b) 4 November, (c) 5 November, 2021 and (d) the quiet day for reference are displayed. The black and gray dots represent the locations of Yanqing and Pingquan, respectively.

Joule heating in polar region during magnetic storm can produce equatorward wind surges and have influence on the MLT region wind field at mid-latitude (Li et al., 2019). However, it is difficult to explain the depletion of metal layers only by the variation of wind field. And the metal layer is electrically neutral and free from the electric field disturbance. During this storm, the atmospheric composition has changed. The data of the thermospheric  $\Sigma O/N_2$  can be accessed from the NASA GUVI Team (2002). Figure 4d shows the average  $\Sigma O/N_2$  in the quiet days of Oct. and Nov. 2021, and there was no obvious  $\Sigma O/N_2$  enhancement region around the world. But in Figs. 4a-4c, the  $\Sigma O/N_2$  enhancement region had extended to mid-latitudes on 3-5 Nov., and on the storm day of 4 Nov., the

Yanqing and Pingquan observation stations had entered the maximum enhancement region as indicated by the yellow, indicating that the thermospheric  $\Sigma O/N_2$  over the two locations increased greatly.

### 4 Discussion

## 4.1 Mechanism for storm time O/N<sub>2</sub> variation

It is difficult to explain the density variations of the neutral metal atoms in the metal layer by only dynamic process, and there was no significant changes in the MLT region wind field as the observation of the meteor radar located in Beijing. The storm time O/N<sub>2</sub> variation as observed by the GUVI is very likely to induce metal layer depletion. Polar temperature increase due to the Joule heating leads to strong upwelling, and thermosphere O and N<sub>2</sub> move upwards with the upwelling (Prölss, 1980, 2011). Vertical advection during the upwelling will reduce O and enhance N<sub>2</sub>, thus leads to the depletion of the O/N<sub>2</sub>. When the downwelling occurs due to the decreasing temperature gradient, the O density increases and N<sub>2</sub> density decreases to enhance the O/N<sub>2</sub> (Yu et al., 2023). Kil et al. (2011) studied the O and N<sub>2</sub> disturbances in F layer during the storm on 20 Nov. 2003. They proposed that the different behaviors of O and N<sub>2</sub> perturbations can't be simply explained by the upwelling and downwelling in the upper atmosphere, and may be related to the global wind circulation, which may transfer the O perturbations from mid-high latitudes to middle-low latitudes, and this transferring process is more remarkable in the low F-region than the high F-region (Mayr et al., 1978; Meier at al., 1995). More importantly, it is necessary to distinguish the changes in oxygen and nitrogen atom density.

## 4.2 Simulation results of the TIEGCM

Figure 5. Simulated global distribution of the column  $O/N_2$  ( $\Sigma O/N_2$ ). Latitude and longitude/ local time maps show the  $\Sigma O/N_2$  variations from 8:20 to 18:20 UT on 4 Nov. 2021 with 2 hour step. The gray curves in each panel are the zero values. The horizontal white dashed lines are the latitude of  $\pm 60^\circ$ . The gray circle is used to indicate the lidar stations [40°N, 116°E].

We apply the TIEGCM to investigate the O and  $N_2$  variation in thermosphere and mesosphere. The simulated longitude-latitude distribution of the percentage changes of the  $\Sigma O/N_2$  from 8:20 to 18:20 UT on 4 Nov. compared to the quiet days is shown in Figure 5. The  $O/N_2$  dataset can be accessed in the Zenodo website (Yu, 2024). In this simulation, the  $O/N_2$  variations are mainly attributed to the equatorward wind disturbances driven by the Joule heating and the influence of Coriolis force is in consideration, the details of the simulated results can refer to Yu et al., (2021; 2023). The horizontal white dashed lines show the latitudes of  $\pm 60^{\circ}$ . The simulated  $\Sigma O/N_2$  over Yanqing and Pingquan is 1.0109, close to the GUVI measurement of 1.0181 on the storm day, and the simulation results are quite accurate. Between 8:20-18:20 UT in Figure 5, the  $\Sigma O/N_2$  over the lidar stations continued increasing. At 8:20 UT in Figure 5a, the increase of the  $\Sigma O/N_2$  near the gray circle didn't exceed 20%,

Figure 6. Simulated oxygen density variations at different altitudes. (a) Storm-quiet time ratio of O and  $N_2$  density between 95-200 km altitude from 5:40 to 20:00 UT on 4 November 2021. Oxygen density variation at (b) 150, 160 and 170 km altitudes, as well as at (c) 100, 102 and 104 km altitudes on the substorm day and reference day.

Both increase of O density and decrease of N<sub>2</sub> density can enhance the O/N<sub>2</sub>. Thus, we investigate the O and N<sub>2</sub> density variation at different heights as shown in Figure 6. Each curve in Fig. 6a shows the height profile of the storm-quiet time percentage of O density (solid curve) and N2 density (dashed curve) between 97-200 km. The quiet time values come from the simulated data on 3 Nov.. The profiles of different time are displayed by different colors. Between 5:40-20:00 UT on 4 Nov., the AE index in Figure 1c reached its maximum and the lidars recorded the lowest metal layer density in this night. As indicated by the height profiles in the same duration in Figure 6a, the enhancement of the O/N2 ratio between 100-200 km altitude was mainly due to the increase of O as well as the decrease of N2. N2 is inert gas and its density variation is unlikely to have a great influence on the density of the metal atoms, thus we mainly focus on the O density variation. Figures 6b and 6c show the O density variations on the storm day compared with on the reference day at lower altitudes. In Figure 6b, the O density enhancements at 150-170 km altitudes were not obvious, and only appeared in a short duration between ~10:00-13:00 UT. However, at the lower heights in Figure 6c, the O density increased even higher and earlier. The enhancement occurred at ~2:30 UT on 4 Nov. and continued to the next day. The beginning of the O enhancement is indicated as a red up arrow in Figure 1e, and the metal layer dissipation should start in the daytime, when the lidars didn't work. The largest increase was 1.9×10<sup>10</sup> atoms cm<sup>-3</sup> at 100 km altitude at 14:00 UT. This increment is much higher than the density of the metal layer. As suggested by previous studies, the storm-time O density enhancement is more in the low F-layer due to the global wind circulation, and the increased oxygen in this region can extend to the E region due to the effect of downwelling (Yu et al., 2023; Kil et al., 2011). Thus, the O increases in lower thermosphere were bound to influence the O variations in the metal layer at an altitude of 80-100 km by molecular diffusion and downwelling, resulting in the O density increase there.

## 4.3 Mechanism for metal layer depletion

Metallic elements are not stable in the Earth's atmosphere, and they will constantly shift between ions, atoms, and compounds. The released Na atoms from meteoroids will be oxidized by O<sub>3</sub> to form NaO (R1) in MLT region, and further react with H<sub>2</sub>O or H<sub>2</sub> to form NaOH (R2 and R3). NaHCO<sub>3</sub> is the recombination of NaOH with CO<sub>2</sub> (R4) and it is the major reservoir for Na on the bottom side of the metal layer (Gómez-Martín et al., 2017; Plane et al., 2015; Yuan et al., 2019). NaHCO<sub>3</sub> is converted back to Na either by photolysis (hv) or by reaction with atom H (R5 & R6). While the O density

increases, more Na atoms will become NaHCO<sub>3</sub> to reduce the Na density.

$$Na + O_3 \rightarrow NaO + O_2 \tag{R1}$$

NaO +
$$H_2O \rightarrow NaOH + OH$$
 (R2)

NaO + H<sub>2</sub>
$$\rightarrow$$
 NaOH + H (R3)

NaOH + CO<sub>2</sub> (+M)
$$\rightarrow$$
 NaHCO<sub>3</sub> (R4)

NaHCO<sub>3</sub> +
$$hv \rightarrow$$
 Na + HCO<sub>3</sub> (R5)

NaHCO<sub>3</sub> + H
$$\rightarrow$$
 Na+ H<sub>2</sub>CO<sub>3</sub> (R6)

Ni atoms in MLT region is oxidized by O<sub>3</sub> and O<sub>2</sub> to form NiO and NiO<sub>2</sub>, respectively (R7 and

R8). These two Ni compounds further react with O3, O2, CO2, and H2O to form higher oxides,

carbonates, and hydroxides (e.g. R9-R14), therein, NiOH, Ni (OH)2 and NiCO3 is the major reservoir

for Ni. These higher compounds will be converted to NiOH and NiO, which are finally converted back

to Ni by reaction with atom H and O, as well as CO (R15-R17) (Daly et al., 2020). The increased O

will convert more Ni atoms into the compound reservoir to dissipate the Ni layer.

321

$$325 \qquad Ni + O_3 \rightarrow NiO + O_2 \tag{R7}$$

$$326 \quad \text{Ni} + \text{O}_2 (+\text{M}) \rightarrow \text{NiO}_2 \tag{R8}$$

$$327 \quad \text{NiO} + \text{O}_3 \rightarrow \text{NiO}_2 + \text{O}_2$$
 (R9)

$$328 \qquad NiO + O_2 \rightarrow ONiO_2 \tag{R10}$$

$$\text{NiO} + \text{CO}_2 (+\text{M}) \rightarrow \text{NiCO}_3$$
 (R11)

NiO + H<sub>2</sub>O (+M)
$$\rightarrow$$
 Ni (OH)<sub>2</sub> (R12)

Ni (OH)<sub>2</sub> + H
$$\rightarrow$$
 NiOH + H<sub>2</sub>O (R13)

$$\text{NiO}_2 + \text{O}_3 \rightarrow \text{ONiO}_2 + \text{O}_2$$
 (R14)

NiOH + H
$$\rightarrow$$
 Ni + H<sub>2</sub>O (R15)

$$334 \qquad NiO + O \rightarrow Ni + O_2 \tag{R16}$$

$$\text{NiO} + \text{CO} \rightarrow \text{Ni} + \text{CO}_2$$
 (R17)

Ca atoms in atmosphere will be converted into the compounds, such as CaO, CaOH, O<sub>2</sub>CaCO<sub>3</sub>, OCaOH, CaCO<sub>3</sub> and so on (e.g. R18-R26). When the O density at the metal layer heights continues to increase, it is likely to promote the formation of O<sub>2</sub>CaOH (R24), which slows the conversion of CaOH to atomic Ca by reacting with atomic H (R27) (Gomez Martin and Plane, 2017). And then O<sub>2</sub>CaOH will convert to the stable compound CaOH (R25 and R28). CaOH will further convert to the stable compound CaCO<sub>3</sub> and enter into a stable chemical cycle as indicated by R24, R29-R31. CaOH and

- CaCO<sub>3</sub> are stable reservoirs for Ca (Plane et al., 2018). As a result, the increased O density converts
- more Ca atoms to the stable compound reservoirs, thus the Ca layer has suffered much severer
- dissipation during the storm as shown in Figure 2.

$$345 Ca + O_3 \rightarrow CaO + O_2 (R18)$$

$$Ca + O_2 (+M) \rightarrow CaO_2$$
 (R19)

$$CaO + CO_2 (+M) \rightarrow CaCO_3$$
 (R20)

$$\operatorname{CaO} + \operatorname{H}_2\operatorname{O} (+\operatorname{M}) \to \operatorname{Ca}(\operatorname{OH})_2$$
 (R21)

$$349 Ca(OH)_2 + H \rightarrow CaOH + H_2O (R22)$$

$$350 \qquad \text{CaCO}_3 + \text{H} \rightarrow \text{CaOH} + \text{CO}_2 \tag{R23}$$

$$CaOH + O_2 (+ M) \rightarrow O_2CaOH (+ M)$$
 (R24)

$$352 \qquad O_2CaOH + O \rightarrow OCaOH + O_2 \tag{R25}$$

$$CaCO_3 + O_2 (+M) \rightarrow O_2 CaCO_3$$
 (R26)

$$354 \qquad \text{CaOH} + \text{H} \rightarrow \text{Ca} + \text{H}_2\text{O} \tag{R27}$$

$$355 \qquad OCaOH + O \rightarrow CaOH + O_2 \tag{R28}$$

$$O_2CaCO_3 + O \rightarrow OCaCO_3 + O_2;$$
 (R29)

$$357 \qquad OCaCO_3 + O \rightarrow CaCO_3 + O_2; \tag{R30}$$

$$CaCO_3 + O_2(+M) \rightarrow O_2CaCO_3;$$
 (R31)

Figure 7. Schematic of the metal layer depletion during a storm. The red circular arrows indicate that high-energy particles have entered the earth. The white transparent area indicates the atmospheric expansion in polar region due to auroral heating, and the yellow arrows show the directions of the heating driven winds with the influence of Coriolis force. The color on the earth indicates the magnitude of the O/N<sub>2</sub>, decreasing in order from red, yellow, green to blue. The translucent dots represent oxygen atoms. Layers of earth's atmosphere are displayed. The red circle around the earth is used to represent the metal layer, and lighter colors indicate lower density.

Figure 7 is drawn to demonstrate the possible mechanism for the metal layer depletion during the super substorm. The high-energy particles enter the earth to enhance Joule heating and increase upper atmosphere temperature (Burns et al., 1995). The temperature enhancements change the horizontal pressure gradient, in conjunction with the ion drag, drive horizontal wind perturbations to modify the global thermospheric circulation. It should be noted that the storm-time horizontal winds are usually influenced by the Coriolis force due to the Earth corotation during the storm evolution (Yu et al., 2022). The storm-induced divergent and convergent winds induce the atmospheric upwelling and downwelling, thus composition changes at high latitudes, which are transported to lower latitudes by the horizontal winds. O increases mainly occur at lower altitudes (Yu et al., 2021), like that around the height of the

metal layer as shown in Figure 6. Metal elements in metal layer cycle between atomic, ionic, and compound states, reaching a conversion equilibrium. The enhancement of O density in the storm has broken this equilibrium, more metal atoms will react with the increased O to produce more metal compounds, thus the density of metal atoms in the metal layer decreased as the lidars observed.

### **5 Conclusions**

Though the density of atmospheric metal layers is varying day to day, the Na, Ca and Ni densities recorded by the Yanqing lidar on the storm day were significantly lower than on other days, and the Na density recorded by the Pingquan lidar was also lowest in the two months. It is quite certain that the super substorm has induced the depletion of the metal layers. The increase of the O density in the MLT region is considered to consume the metal atoms.

Magnetic storms affect the ionosphere and thermosphere by disturbing the electric field and wind field usually. Metal layers are located at a lower altitude, where the ambient atmospheric density is much greater. It is generally believed that the influence of magnetic storms is difficult to reach the altitudes of metal layer. Our observations present two very interesting phenomena, one is that the effects of magnetic storm have penetrated through the thermosphere and reached the MLT region, and the other is that the pathway of storm influence is no longer the usual electrodynamic process, but a chemical process. The storm has provided us a very nice opportunity to record the responses of the metal layer to breaking of the chemical equilibrium in MLT region. It implies that magnetic storms have a broader impact on Earth's atmosphere, and a new horizon is opened up for us to study of the impact of magnetic storms on the Earth's space environment.

This observation was carried out only at mid-latitudes of China. However, what happens at other latitudes is unknown, especially more significant impact may occur at higher latitudes as Figure 4 indicated. Thus, it is necessary to record the storm influence on mesospheric metal layers at different latitudes for more latitude coverage. The new observation station in Mohe, Heilongjiang province, China (122.37°E, 53.5°N) is now in operation. We will report more observation results at higher latitude in the coming years.

Data availability. The data of metal atoms can be publicly obtained in the Zenodo website

(https://zenodo.org/records/15003984) (Xu and Chen, 2025). The geomagnetic indices can be freely 404 accessed in the OMNIWeb (1963) (https://omniweb.gsfc.nasa.gov/). TIMED GUVI observations of 405 O/N<sub>2</sub> column density ratios are available from <a href="https://guvitimed.jhuapl.edu/">https://guvitimed.jhuapl.edu/</a> (NASA GUVI Team, 2002). 406 The TIE-GCM simulation datasets supporting this study are available in the Zenodo website 407 (https://zenodo.org/records/14232871) (Yu, 2024). 408 Author contributions. Y.X. wrote the manuscript and G. C. rewrote it. Y. X. conducted comprehensive 409 data analysis. G. C. investigated the physicochemical mechanisms. G. Y., P. H., and M. Z. provided 410 helpful suggestions and revised the manuscript. Z. R. and T. Y. provided TIEGCM simulation results. F. 411 W. assisted in nickel atoms data processing. G. Y., L. D., H. Z., X. C., and F. L. organized the datasets. 412 All the authors discussed the results and commented on the manuscript. 413 Competing interests. The contact author has declared that none of the authors has any competing 414 interests. 415 Acknowledgements. This project was supported by the National Natural Science Foundation of China 416 (42274197 and 41722404). We acknowledge the use of the lidar data from the Chinese Meridian 417 Project. 418 Reference 419 Burns, A. G., Killeen, T. L., Deng, W., Carignan, G. R., & Roble, R. G. Geomagnetic storm effects in 420 the low- to middle-latitude upper thermosphere. Journal of Geophysical Research, 100(A8), 421 14673. https://doi.org/10.1029/94ja03232, 1995 422 Chen, G. et al.: Anomalous sporadic-E enhancements and field-aligned irregularities at low-latitudes 423 during the intense geomagnetic activities on 17–18 March 2015. Journal of Geophysical Research: 424 Space Physics, 128(10), e2023JA031856. https://doi.org/10.1029/2023JA031856, 2023. 425 Christensen, A. B. et al.: Initial observations with the Global Ultraviolet Imager (GUVI) in the NASA 426 TIMED satellite mission. Journal of Geophysical Research: Space Physics, 108(A12), 1,451.

https://doi.org/10.1029/2003JA009918, 2003.

- Chu, X. and Papen, G. C.: Resonance fluorescence lidar for measurements of the middle and upper
- atmosphere, in Laser Remote Sensing, Fujii, T. and Fukuchi, T., eds. (CRC Press), 179-432, 2005.
- Daly, S. M., Feng, W., Mangan, T. P., Gerding, M., & Plane, J. M. C.: The Meteoric Ni Layer in the
- Upper Atmosphere. Journal of Geophysical Research: Space Physics, 125, e2020JA028083.
- <a href="https://doi.org/10.1029/2020JA028083">https://doi.org/10.1029/2020JA028083</a>, 2020.
- Du, L. F. et al.: Continuous detection of diurnal sodium fluorescent lidar over Beijing in China.
- Atmosphere, 11(1), 118. doi:10.3390/atmos11010118, 2020.
- Du, L. F. et al.: The all-solid-state narrowband lidar developed by optical parametric
- oscillator/amplifier (OPO/OPA) technology for simultaneous detection of the Ca and Ca<sup>+</sup> layers.
- Remote Sensing, 15(18), 4,566. doi:10.3390/rs15184566, 2023.
- Eska, V., Höffner, J. and von Zahn, U.: Upper atmosphere potassium layer and its seasonal variations at
- 54°N. Journal of Geophysical Research: Space Physics, 103(A12), 29,207–29,214.
- <u>https://doi.org/10.1029/98JA02481</u>, 1998.
- Gao, Q., Chu, X., Xue, X., Dou, X., Chen, T., and Chen, J., Lidar observations of thermospheric Na
- layers up to 170 km with a descending tidal phase at Lijiang (26.7°N, 100.0°E), China, J. Geophys.
- Res. Space Physics, 120, 9213-9220, doi:10.1002/2015JA021808, 2015.
- Gardner, C. S. et al.: Seasonal variations of the Na and Fe layers at the south pole and their implications
- for the chemistry and general circulation of the polar mesosphere. Journal of Geophysical
- Research: Atmospheres, 110(D10), D10302. <a href="https://doi.org/10.1029/2004JD005670">https://doi.org/10.1029/2004JD005670</a>, 2005.
- Gardner, C. S. et al.: Seasonal variations of the mesospheric Fe layer at Rothera, Antarctica (67.5°S,
- 68.0°W). Journal of Geophysical Research: Atmospheres, 116(D2), D02304.
- https://doi.org/10.1029/2010JD014655, 2011.
- Gerding, M., Alpers, M., von Zahn, U., Rollason, R. J. and Plane, J. M. C.: Atmospheric Ca and Ca<sup>+</sup>
- layers: Midlatitude observations and modeling. Journal of Geophysical Research: Space Physics,
- 105(A12), 27,131–27,146. https://doi.org/10.1029/2000JA900088, 2000.
- Gerding, M., Daly, S., and Plane, J.M.C.. Lidar Soundings of the Mesospheric Nickel Layer Using
- Ni(3F) and Ni(3D) Transitions. Geophys. Res. Lett. 46. doi:10.1029/2018GL080701, 2018.
- Gómez Martin, J. C. and Plane, J. M. C.: Reaction kinetics of CaOH with H and O2 and O2CaOH with
- O: Implications for the atmospheric chemistry of meteoric calcium. ACS Earth and Space
- Chemistry, 1(7), 431-441. <a href="https://doi.org/10.1021/acsearthspacechem.7b00072">https://doi.org/10.1021/acsearthspacechem.7b00072</a>, 2017.

- Gong, S. H. et al.: Lidar studies on the nighttime and seasonal variations of background sodium layer at
- different latitudes in China. Chinese Journal of Geophysics, 56(8), 2511-2521.
- doi:10.6038/cjg20130802, 2013.
- Hagan, M. E., and Forbes, J. M.: Migrating and nonmigrating diurnal tides in the middle and upper
- atmosphere excited by tropospheric latent heat release. Journal of Geophysical Research:
- Atmospheres, 107(D24), 4,754. <a href="https://doi.org/10.1029/2001JD001236">https://doi.org/10.1029/2001JD001236</a>, 2002.
- Hagan, M. E., and Forbes, J. M.: Migrating and nonmigrating semidiurnal tides in the upper
- atmosphere excited by tropospheric latent heat release. Journal of Geophysical Research: Space
- Physics, 108(A2), 1,062. https://doi.org/10.1029/2002JA009466, 2003.
- Heelis, R. A., Lowell, J. K. and Spiro, R. W.: A model of the high-latitude ionospheric convection
- pattern. Journal of Geophysical Research: Space Physics, 87(A8), 6,339–6,345.
- <u>https://doi.org/10.1029/JA087iA08p06339</u>, 1982.
- Hickey, M. P. and Plane, J. M. C.: A chemical-dynamical model of wave-driven sodium fluctuations.
- Geophysical Research Letters, 22(20), 2861-2864. https://doi.org/10.1029/95GL02784, 1995.
- Hunten, D.M.: Spectroscopic studies of the twilight airglow. Space Science Reviews 6, 493-573.
- https://doi.org/10.1007/BF00173704, 1967.
- Jiao, J. et al.: First report of sporadic K layers and comparison with sporadic Na layers at Beijing,
- China (40.6°N, 116.2°E). Journal of Geophysical Research: Space Physics, 120(6), 5,214–5,225.
- <a href="https://doi.org/10.1002/2014JA020955">https://doi.org/10.1002/2014JA020955</a>, 2015.
- Jiao, J. et al.: A Comparison of the midlatitude nickel and sodium layers in the mesosphere:
- Observations and modeling. Journal of Geophysical Research: Space Physics, 127(2),
- e2021JA030170. https://doi.org/10.1029/2021JA030170, 2022.
- Kil, H., Kwak, Y.-S., Paxton, L. J., Meier, R. R. and Zhang Y.: O and N<sub>2</sub> disturbances in the F region
- during the 20 November 2003 storm seen from TIMED/GUVI. Journal of Geophysical Research:
- Space Physics, 116(A2), A02314. https://doi.org/10.1029/2010JA016227, 2011.
- Kopp, E.: On the abundance of metal ions in the lower ionosphere. Journal of Geophysical Research:
- Space Physics, 102(A5), 9667–9674. https://doi.org/10.1029/97JA00384, 1997.
- Li, J. Y. et al.: A modeling study of the responses of mesosphere and lower thermosphere winds to
- geomagnetic storms at middle latitudes. Journal of Geophysical Research: Space Physics, 124(5),
- 3,666–3,680. <a href="https://doi.org/10.1029/2019JA026533">https://doi.org/10.1029/2019JA026533</a>, 2019.

- Li, Y. X. et al.: Observational evidence for the neutral wind responses in the mid-latitude lower
- thermosphere to the strong geomagnetic activity. Space Weather, 22(9), e2023SW003830.
- https://doi.org/10.1029/2023SW003830, 2024.
- Mayr, H. G., Harris, I. and Spencer, N. W.: Some properties of upper atmosphere dynamics. Reviews of
- Geophysics, 16(4), 539–565. https://doi.org/10.1029/RG016i004p00539, 1978.
- Megie, G., Bos, F., Blamont, J. and Chanin, M.: Simultaneous nighttime lidar measurements of
- atmospheric sodium and potassium, Planet. Space Sci. 26, 27-35,
- <u>https://doi.org/10.1016/0032-0633(78)90034-X</u>, 1978.
- Meier, R. R., Cox, R., Strickland, D. J., Craven, J. D. and Frank L. A.: Interpretation of dynamics
- explorer far UV images of the quiet time thermosphere. Journal of Geophysical Research: Space
- Physics, 100(A4), 5,777–5,794. https://doi.org/10.1029/94JA02679, 1995.
- NASA GUVI Team.: GUVI level 3 data products [Dataset]. Thermospheric O/N2.
- <a href="https://guvitimed.jhuapl.edu/">https://guvitimed.jhuapl.edu/</a>, 2002.
- OMNIWeb.: Geomagnetic indices [Dataset]. OMNIWeb. Retrieved from
- https://omniweb.gsfc.nasa.gov/, 1963.
- Picone, J. M., Hedin, A. E., Drob, D. P., and Aikin, A. C.: NRLMSISE-00 empirical model of the
- atmosphere: Statistical comparison and scientific issues. J. Geophys. Res. 107(A12): 1468.
- doi:10.1029/2002JA009430, 2002.
- Plane, J. M. C., Feng, W. and Dawkins, E. C. M.: The mesosphere and metals: Chemistry and changes.
- Chemical Reviews, 115(10), 4497-4541. https://doi.org/10.1021/cr500501m, 2015.
- Plane, J. M. C., Feng, W. H., Gómez Martín, J. C., Gerding, M., and Raizada, S.: A new model of
- meteoric calcium in the mesosphere and lower thermosphere. Atmospheric Chemistry and Physics,
- 18(20), 14,799–14,811. https://doi.org/10.5194/acp-18-14799-2018, 2018.
- Prölss, G. W.: Magnetic storm associated perturbations of the upper atmosphere: Recent results
- obtained by satellite-borne gas analyzers. Reviews of Geophysics, 18(1), 183–202.
- <u>https://doi.org/10.1029/RG018i001p00183</u>, 1980.
- Prölss, G. W.: Density perturbations in the upper atmosphere caused by the dissipation of solar wind
- energy. Surveys in Geophysics, 32, 101–195. <a href="https://doi.org/10.1007/s10712-010-9104-0">https://doi.org/10.1007/s10712-010-9104-0</a>, 2011.
- Raizada, S., and Tepley, C. A.: Seasonal variation of mesospheric iron layers at Arecibo: First results
- from low-latitudes. Geophysical Research Letters, 30(2), 1082.

- https://doi.org/10.1029/2002GL016537, 2003.
- Resende, L. C. A., Shi, J. K., Denardini, C. M., Batista, I. S., Nogueira, P. A. B., Arras, C., et al.: The
- influence of disturbance dynamo electric field in the formation of strong sporadic E layers over
- Boa Vista, a low-latitude station in the American sector. Journal of Geophysical Research: Space
- Physics, 125(7), e2019JA027519. https://doi.org/10.1029/2019JA027519, 2020.
- Richmond, A. D., Ridley, E. C., and Roble, R. G.: A thermosphere/ionosphere general circulation
- model with coupled electrodynamics. Geophysical Research Letters, 19(6), 601-604.
- <u>https://doi.org/10.1029/92GL00401</u>, 1992.
- Roble, R. G., and Ridley, E. C.: An auroral model for the NCAR thermosphere general circulation
- model (TGCM). Annales Geophysicae, 5, 369–382., 1987.
- Roble, R. G., Ridley, E. C., Richmond, A. D., and Dickinson, R. E.: A coupled
- thermosphere/ionosphere general circulation model. Geophysical Research Letters, 15(12),
- 1,325-1,328. https://doi.org/10.1029/GL015i012p01325, 1988.
- Solomon, S. C., and Qian, L.: Solar extreme-ultraviolet irradiance for general circulation models.
- Journal of Geophysical Research: Space Physics, 110(A10), A10306.
- https://doi.org/10.1029/2005JA011160, 2005.
- Welsh, B. and Gardner C.: Nonlinear resonant absorption effects on the design of resonance
- fluorescence lidars and laser guide stars, Appl. Opt., 28, 4141-4153,
- <u>https://doi.org/10.1364/AO.28.004141</u>, 1989.
- Wu, F., Zheng, H., Cheng, X., Yang, Y., Li, F., Gong, S., Du, L., Wang, J., Yang, G., Simultaneous
- detection of the Ca and Ca<sup>+</sup> layers by a dual-wavelength tunable lidar system, Appl Opt., 59(13),
- 4122-4130, doi: 10.1364/AO.381699, 2020.
- Wu, F. J. et al.: Lidar observations of the upper atmospheric nickel layer at Beijing (40°N,116°E).
- Journal of Quantitative Spectroscopy and Radiative Transfer, 260, 107, 468.
- <u>https://doi.org/10.1016/j.jqsrt.2020.107468</u>, 2021.
- Wu, F., Chu, X., Du, L., Jiao, J., Zheng, H., Xun, Y., et al.: First simultaneous lidar observations of
- thermosphere-ionosphere sporadic Ni and Na (TISNi and TISNa) layers (~105-120 km) over
- Beijing (40.42°N, 116.02°E). Geophysical Research Letters, 49, e2022GL100397.
- <u>https://doi.org/10.1029/2022GL100397</u>, 2022.
- Xu, Y. M., and Chen, G.: Metal atoms dataset [Dataset]. Zenodo. https://zenodo.org/records/15003984,

- 2025.
- Xun, Y. et al.: Seasonal variation in the mesospheric Ca layer and Ca<sup>+</sup> layer simultaneously observed
- over Beijing (40.41°N, 116.01°E). Remote Sensing, 16(3), 596. doi:10.3390/rs16030596, 2024.
- Yi, F. et al.: Seasonal variations of the nocturnal mesospheric Na and Fe layers at 30°N. Journal of
- Geophysical Research: Atmospheres, 114(D1), D01301. https://doi.org/10.1029/2008JD010344,
- 2009.
- Yu, T., Wang, W., Ren, Z., Cai, X., Yue, X., & He, M.. The response of middle thermosphere (~ 160
- 555 km) composition to the 20-21 November 2003 superstorm, Journal of Geophysical Research:
- Space Physics, 126, e2021JA029449, https://doi.org/10.1029/2021ja029449, 2021.
- Yu, T., Cai, X., Ren, Z., Li, S., Pedatella, N., & He, M.. Investigation of the ΣO/N2 depletion with
- latitudinally tilted equatorward boundary observed by GOLD during the geomagnetic storm on
- April 20, 2020. Journal of Geophysical Research: Space Physics, 127, e2022JA030889.
- https://doi. org/10.1029/2022JA030889, 2022.
- Yu, T., Cai, X., Ren, Z., Li, S., Pedatella, N., & He, M.. Investigation of the ΣO/N2 depletion with
- latitudinally tilted equatorward boundary observed by GOLD during the geomagnetic storm on
- April 20, 2020. Journal of Geophysical Research: Space Physics, 127, e2022JA030889. https://doi.
- org/10.1029/2022JA030889, 2022.Yu, T. T. et al.: Investigation of interhemispheric asymmetry of
- the thermospheric composition observed by GOLD during the first strong geomagnetic storm in
- solar-cycle 25, 1: IMF By effects. Journal of Geophysical Research: Space Physics, 128(10),
- 67 e2023JA031429. https://doi.org/10.1029/2023JA031429, 2023.
- Yu, T. T.: TIEGCM 2019\_11 storm [Dataset]. Zenodo. <a href="https://zenodo.org/records/14232871">https://zenodo.org/records/14232871</a>, 2024.
- Yuan, T., Feng, W., Plane, J. M. C., and Marsh, D. R.: Photochemistry on the bottom side of the
- mesospheric Na layer, Atmos. Chem. Phys., 19, 3769-3777,
- https://doi.org/10.5194/acp-19-3769-2019, 2019.
- Yue, X. C., Friedman, J. S., Wu, X. B. and Zhou Q. H.: Structure and seasonal variations of the
- nocturnal mesospheric K layer at Arecibo. Journal of Geophysical Research: Atmospheres,
- 122(14), 7,260–7,275. https://doi.org/10.1002/2017JD026541, 2017.
- Zhang, Y. et al.: O/N2 changes during 1-4 October 2002 storms: IMAGE SI-13 and TIMED/GUVI
- observations. Journal of Geophysical Research: Space Physics, 109(A10), A10308.
- <u>https://doi.org/10.1029/2004JA010441, 2004.</u>