# Peer review of "Metal Layer Depletion during the Super Substorm on 4 1"

_EGUsphere, 2025_

## Referee Comment (RC1)

**Review of ACP Manuscript egusphere-2025-1888**

The authors describe the depletion of the meteoric Na, Ni, and Ca layers observed with lidars at several mid-latitude sites in eastern China during a Super Substorm in Nov 2021. The paper is well-organized, clearly written, and certainly of interest to the upper atmosphere science community. The metal layer depletion, coincident with the substorm, is unequivocal and the authors provide a well-reasoned description of a chemical mechanism that is plausibly responsible for the depletions. The paper is easy to read, logically organized, and shows convincingly that the depletions are unusual and highly correlated with standard indices used to characterize geomagnetic activity.

I recommend the paper be accepted for publication in its current form, subject to some minor editorial corrections and suggestions listed below.

Line 33        typo "field"

Line 46        typo "winter"

Line 55        typo "geomagnetic"

Figure 2        this is an important figure that contains a large amount of relevant information, but in its current form it is hard read because the panels and text are small. I suggest rotating the figure 180˚ to make it a 4x5 figure (which could be expanded to a full page), and perhaps consider using a color scale for the density contours. Alternatively, perhaps the authors could prepare separate figures for the different metals, although there is value in including all the metals in a single figure.

Lines 133-140        change the density units from "particles/cm$^3$" to "atoms/cm$^3$"

Figure 3        I could not see the error bars. Perhaps it is sufficient to just quote an upper bound such less than x% in the caption.

Lines 172-176        I found this sentence confusing and was especially surprised at the very low cross-correlation between the Pingquan Na and Yanqing Ca abundances. I wonder if this calculation should just be eliminated because Figs. 2 and 3 clearly show all measurements at the lidar sites showed significant depletion of the metals during the substorm. While the cross-correlation coefficients are interesting, the values may raise more questions than they answer.

Line 196        "there were no significant changes in the MLT…"

---

## Referee Comment (RC2)

**Reviewer Comments to Author(s):**

The manuscript "Metal Layer Depletion during the Super Substorm on 4 November 2021" presents intriguing observations of metal layer depletion during a super substorm, combining lidar data, GUVI satellite observations, and numerical simulation. The integration of multi-instrument data and modelling to link metal layer changes with geomagnetic storm-induced oxygen density variations is a strong point. However, several issues related to clarity, grammar, scientific argumentation, and figure presentation need attention to enhance the paper's impact and precision.

**Line 17-18:**

"Due to the viscous force of air, the wind and electric field disturbances of a magnetic storm is hard to penetrate deep into the Earth's dense atmospheric region."

The statement is oversimplified and lacks precise backing. While viscous forces and neutral atmosphere density limit electric field penetration, wave and particle interactions can cause storm effects in the MLT region. Clarify referencing and nuances.

**Line 20-26:**

The core hypothesis that metal atom depletion is caused by increased atomic oxygen forming metal compounds during storm is compelling. However, more direct chemical kinetic evidence or references supporting the rates and abundances of such reactions at MLT altitudes during storms would strengthen this argument.

Suggest discussing or referencing existing atmospheric metal chemistry models more explicitly.

**Line 42-43:**

"It is widely accepted that chemical reactions below 85 km altitude have an important impact on the density variation of Na atoms, but the chemical reaction above 85 km has little influence."

This conflicts with later claims about chemical reactions influencing metal layers during storms at ~90-100 km. The authors should clarify how their findings contrast with or extend prior understanding.

**Line 55-58:**

Claiming that the storm cannot penetrate to metal layers is common but somewhat oversimplified. Some studies have shown ion-neutral coupling effects in MLT during storms — this could be acknowledged.

**Line 58-59:**

"Whether a storm can affect the atmospheric layers as low as the Mesosphere and Low Thermosphere (MLT) region is a very interesting topic."

This is true, but not a novel question. The authors should better situate their work in the context of previous MLT storm effects research.

**Line 64:**

Statement that this is the first time a storm effect on metal layer has been introduced is somewhat strong. The work is likely first strong evidence of depletion caused by changed chemistry, but literature on sporadic E-layer and metal ion responses exists. This should be framed carefully.

**Line 69-76:**

Description of lidar systems needs clarity on detection limits, uncertainty, and calibration methods. Are all metal densities quantitative and directly comparable?

**Line 94-113 (Geomagnetic Indices):**

The storm description is adequate. However, consider adding direct linkage of timing of substorm peaks with metal layer depletion timing for clarity.

**Line 167-176:**

The statistical argument that simultaneous depletion at 4 stations with low chance of random occurrence ($p=1.334\times10^{-7}$) supports storm causality is good.

However, the low correlation coefficients between metal layers (e.g. $3.578\times10^{-4}$ for Pingquan-Na and Yanqing-Ca) invite discussion — why do metal species respond differently? Is variability noise or physical reason? This warrants explanation.

**Line 209-250:**

The TIEGCM model description and simulated O/N2 enhancements support the scenario. It is a limitation that TIEGCM bottom boundary is ~97 km, slightly above some metal layers' lower height ~80 km. Authors should discuss how simulation resolution affects conclusions and chemical coupling.

**Line 252-291:**

The detailed chemical pathway for Ca compounds affecting Ca atom abundance is well presented (Equations 1-6).

For a stronger theoretical argument, extend these to Na, Ni chemistry or provide reasoning why Ca is emphasized.

**Line 293-312:**

Recommend highlighting limitations explicitly (measurement spatial coverage limited to mid-latitudes China, need for more latitude coverage).

Suggest stating potential implications on atmospheric chemistry and space weather forecasting more explicitly.

**3. Grammatical, Syntax, and Stylistic Issues**

**Line 17:** "wind and electric field disturbances of a magnetic storm is hard" → "are hard" (plural subject-verb agreement)

**Line 20:** "metal layers were observed to deplete by three lidars" → "were observed to deplete by three lidars" is unclear. Better: "were observed to decrease by observations from three lidars"

**Line 21:** "The Na , Ca and Ni densities on the storm day" → remove extra space after Na

**Line 22:** "The O/N2 column density ratio ... was much higher than that on the quiet days," → "was much higher than on quiet days,"

**Line 35:** "turn into compounds, deposit on the Earth's" → "turn into compounds, depositing..." / or "turn into compounds that deposit..."

**Line 40:** "density of Na peaks in winter and reaches its lowest level in summer at all latitudes" → Add comma after "winter"

**Line 56:** "variations in ionosphere," → "variations in the ionosphere,"

**Line 64:** "and it is the first time to introduce the influence of storm" → "and this is the first time the influence of a storm on the metal layer is introduced"

**Line 70:** "dual-wavelength simultaneous detection system is used for Na and Ni observation and the all-solid-state narrowband lidar is used to observe the Ca layer" → awkward; better split into two sentences

**Line 102:** "20:00 UT indicating beginning of the storm" → "indicating the beginning of the storm"

**Line 124:** "metal atoms dataset can be accessed in" → "metal atoms dataset is accessible at"

**Line 138-140:** Multiple phrasing issues with awkward expressions like "is lower than those on the reference days as well as those of month average" → "is lower than on reference days and monthly averages"

**Line 169:** "but the lowest column abundances in all the four plots of Fig. 3 occurred on 4 Nov. 2021 as shown by the red bars." → "in all four plots of Fig. 3" (remove "the")

**Line 200:** "vertical advection during the upwelling will reduce O and enhance $N_2$, thus lead to the depletion of the $O/N_2$." → "thus leads to the depletion"

**Line 221:** "The simulated $\Sigma O/N_2$ over Yanqing and Pingquan is 1.0109 and the GUVI measurement is 1.0181..." → The values are very close; it would be better to say "is 1.0109, close to the GUVI measurement of 1.0181"

**Line 246:** "The largest increase was $1.9\times10^{10}/cm^3$ at 100 km altitude..." → write as "$1.9 \times 10^{10}$ cm$^{-3}$" with correct spacing and units

**Line 262:** "As a result, the increased O density converts more Ca atoms to compounds, and thus the Ca density in the metal layer decreases significantly." → merge last two clauses: "...to compounds, thus decreasing Ca density significantly."

---

## Author Comment (AC1)

**Response to Reviewer #1**

1、 The authors describe the depletion of the meteoric Na, Ni, and Ca layers observed with lidars at several mid-latitude sites in eastern China during a Super Substorm in Nov 2021. The paper is well-organized, clearly written, and certainly of interest to the upper atmosphere science community. The metal layer depletion, coincident with the substorm, is unequivocal and the authors provide a well-reasoned description of a chemical mechanism that is plausibly responsible for the depletions. The paper is easy to read, logically organized, and shows convincingly that the depletions are unusual and highly correlated with standard indices used to characterize geomagnetic activity.

I recommend the paper be accepted for publication in its current form, subject to some minor editorial corrections and suggestions listed below.

**Response:**

We appreciate your thorough evaluation and constructive feedback on our manuscript. Your recognition of the organization, clarity, and scientific merit of our manuscript is highly encouraging. We have carefully addressed the minor editorial corrections and suggestions, hereby provide a detailed account of the revisions, and the manuscript has been revised according to all of your suggestions and comments.

Thank you again for your valuable input, which has significantly improved the manuscript's rigor and readability. We believe the revisions address all concerns and we hope this revised manuscript can make you satisfied.

2、 Line 33  typo "field"

3、 Line 46  typo "winter"

4、 Line 55  typo "geomagnetic"

**Response:**

We are grateful for your careful reading and for identifying the typographical errors in our

manuscript. All noted mistakes have been corrected, and we have performed additional proofreading to enhance the overall quality of the text.

" wind filed" is revised to " wind field".

" the density peak in winder" is revised to " the density peak in winter".

" geomengnetic storm on the earth" is revised to " geomagnetic storm on the earth".

5、Figure 2 this is an important figure that contains a large amount of relevant information, but in its current form it is hard read because the panels and text are small. I suggest rotating the figure 180°to make it a 4x5 figure (which could be expanded to a full page), and perhaps consider using a color scale for the density contours. Alternatively, perhaps the authors could prepare separate figures for the different metals, although there is value in including all the metals in a single figure.

**Response:**

Thank you so much for your valuable feedback. We have carefully considered your suggestions and implemented the following revisions.

This figure is rotated 180° to make it consist of 4×5 subplots, and we enlarge this figure to make it occupy an entire page. The text in the figure has also been enlarged by two font sizes. The first version of the figure is coloured. To make the figure more friendly for people with color vision deficiencies, we used grayscale images. And we also find that the grayscale images can present more details of the metal layer variations. Presenting four sets of data in four separate figures can not provide readers with an intuitive understanding of the metal layer depletion. Different data with common features will have a better display effect when presented in one figure.

[Figure]

**Figure 2. Lidar observations of the metal layers. The lidar observations of the Na density in Pingquan, as well as the Na, Ca and Ni density in Yanqing are listed from top to bottom row. The observations on the substorm day of 4 Nov. 2021 are displayed in the third column and highlighted by the red dashed box. The observations on the reference days are shown in the second and forth columns. The monthly average of November is shown in the fifth column. The first column shows the altitude profiles of the average metal density for a day, and the error bar indicates the uncertainty in density measurements. LT = UT + 8.**

6、Lines 133-140 change the density units from "particles/cm$^3$" to "atoms/cm$^3$"

**Response:**

We sincerely appreciate your valuable suggestion. We also consider the suggestion of Reviewer #2.

All the unit "particles/cm$^3$" in the manuscript are revised to "atoms·cm$^{-3}$".

7、Figure 3 I could not see the error bars. Perhaps it is sufficient to just quote an upper bound such less than x% in the caption.

**Response:**

Thanks so much for your suggestion. We have added the column abundance on the top of each bar in Figure 3.

[Figure]

**Figure 3. Average column abundances of the metal layers. The observations of the (a) Na lidar in Pingquan, as well as the (b) Na lidar, (c) Ca lidar and (d) Ni lidar in Yanqing are listed from top to bottom row. The red bars indicate the observations on the substorm day. Each vertical bar shows the average column abundance between 75-120 km altitude in one night and all the valid observation results in October and November 2021 are displayed. The uncertainty of the column abundance is exhibited on the top of each bar.**

8、Lines 172-176 I found this sentence confusing and was especially surprised at the very low cross-correlation between the Pingquan Na and Yanqing Ca abundances. I wonder if this calculation should just be eliminated because Figs. 2 and 3 clearly show all measurements at the lidar sites showed significant depletion of the metals during the substorm. While the cross-correlation coefficients are interesting, the values may raise more questions than they answer.

**Response:**

We sincerely thank you for this helpful suggestion, which has improved our manuscript significantly.

As you anticipated, the second reviewer also raised the issue about the cross-correlation coefficients. Thus, we can no longer simply avoid this problem by just removing the corresponding context. We will answer the reviewers as best as we can.

Table 1. Cross-correlation coefficients of different data

| Item | Pingquan Na-Yanqing Na | Pingquan Na-Yanqing Ca | Pingquan Na-Yanqing Ni | Yanqing Na-Ca | Yanqing Na-Ni | Yanqing Ca-Ni |
|---|---|---|---|---|---|---|
| Correlation coefficient | 0.397 | $3.578 \times 10^{-4}$ | 0.507 | 0.346 | 0.603 | 0.169 |

All the correlation coefficients of the metal atom abundance variations are relatively low, indicating there are no much relationship between them. However, this does not mean that they are not correlated over a longer time scale, such as one year (Höffner and Friedman, 2004). We also can find that the abundance variation of the Ca has presented less correlation with other atoms, and the correlation coefficient of Pingquan Na-Yanqing Ca is only $3.578 \times 10^{-4}$. Observations indicate that the Ni and Na layers show close correlations on the scale of hours (Wu et al., 2022). And the calcium layer is somewhat unique. It find that although Ca abundance has a similar elemental abundance to Na in meteorites, the Ca atom abundance in metal layer is roughly 2 orders of magnitude smaller than Na. Plane et al., (2018) suggested that CaOH and $CaCO_3$ are stable reservoirs for Ca in metal layer, as a result, more Ca atoms are converted to compound and the Ca atom abundance in metal layer is much less than the Na atom abundance. The Ca abundance variation is more affected by chemical reactions, so it has less correlation with other metal atoms. The relevance between the Ca and Na in different places further decrease, thus the estimated correlation coefficient is very low.

This is our speculation based on the limited observations. It is not the focus of this manuscript and we do not intend to discuss more about this topic in this manuscript.

**Reference**

Höffner, J. and Friedman, J. S.: The mesospheric metal layer topside: a possible connection to meteoroids. Atmos. Chem. Phys., 4, 801-808. https://doi.org/10.5194/acp-4-801-2004, 2004.

Plane, J. M. C., Feng, W. H., Gómez Martín, J. C., Gerding, M., and Raizada, S.: A new model of meteoric calcium in the mesosphere and lower thermosphere. Atmospheric Chemistry and Physics, 18(20), 14,799–14,811. https://doi.org/10.5194/acp-18-14799-2018, 2018.

Wu, F., Chu, X., Du, L., Jiao, J., Zheng, H., Xun, Y., et al.: First simultaneous lidar observations of thermosphere-ionosphere sporadic Ni and Na (TISNi and TISNa) layers (~105-120 km) over Beijing (40.42°N, 116.02°E). Geophysical Research Letters, 49, e2022GL100397. https://doi.org/10.1029/2022GL100397, 2022.

9、Line 196 "there were no significant changes in the MLT…"

**Response:**

We greatly appreciate your helpful recommendations.

"there was no much changes in the MLT region wind field..." is revised to "there were no significant changes in the MLT region wind field..."

---

## Author Comment (AC2)

**Response to Reviewer # 2**

**Reviewer Comments to Author(s):**

The manuscript "Metal Layer Depletion during the Super Substorm on 4 November 2021" presents intriguing observations of metal layer depletion during a super substorm, combining lidar data, GUVI satellite observations, and numerical simulation. The integration of multi-instrument data and modelling to link metal layer changes with geomagnetic storm-induced oxygen density variations is a strong point. However, several issues related to clarity, grammar, scientific argumentation, and figure presentation need attention to enhance the paper's impact and precision.

**Response:**

We greatly appreciate the your positive assessment of the multi-instrument and modeling approach of our study. We fully accept your suggestions and comments. We have thoroughly revised the manuscript to address these concerns, which include: A comprehensive proofreading to correct grammatical errors and improve sentence clarity throughout the text. A restructuring of several key arguments to enhance their logical flow and scientific precision, particularly in the Introduction and Discussion sections. According to your helpful suggestions and comments, the grammar, scientific argumentation, and figure presentation of this manuscript have been improved greatly. Detailed responses regarding specific figures are provided below.

We believe these revisions have substantially strengthened the manuscript, and we are grateful for your guidance.

1. **Line 17-18:**

"Due to the viscous force of air, the wind and electric field disturbances of a magnetic storm is hard to penetrate deep into the Earth's dense atmospheric region."

The statement is oversimplified and lacks precise backing. While viscous forces and neutral atmosphere density limit electric field penetration, wave and particle interactions can cause storm effects in the MLT region. Clarify referencing and nuances.

**Response:**

Thanks so much for your carefulness and valuable suggestions.

Whether a storm can affect the atmospheric layers as low as the Mesosphere and Low Thermosphere (MLT) region is a very interesting topic. In this height range, a number of studies on the wind field disturbances and variations in ionospheric E-region have been conducted (e.g. Chen et al., 2023; Li et al., 2024). Our expression is not rigorous. To avoid misunderstanding by the readers, we have revised this sentence as following.

There are some studies about the wind field disturbances in Mesosphere and Low Thermosphere (MLT) region and the plasma variations in ionospheric E-region during magnetic storms, but no study on the impact of storms on the metal atoms in mesosphere. During the super substorm on 4 Nov. 2021...

**Reference**

Chen, G. et al.: Anomalous sporadic-E enhancements and field-aligned irregularities at low-latitudes during the intense geomagnetic activities on 17–18 March 2015. Journal of Geophysical Research: Space Physics, 128(10), e2023JA031856. https://doi.org/10.1029/2023JA031856, 2023.

Li, Y. X. et al.: Observational evidence for the neutral wind responses in the mid-latitude lower thermosphere to the strong geomagnetic activity. Space Weather, 22(9), e2023SW003830. https://doi.org/10.1029/2023SW003830, 2024.

2. **Line 20-26:**

The core hypothesis that metal atom depletion is caused by increased atomic oxygen forming metal compounds during storm is compelling. However, more direct chemical kinetic evidence or references supporting the rates and abundances of such reactions at MLT altitudes during storms would strengthen this argument.

Suggest discussing or referencing existing atmospheric metal chemistry models more explicitly.

**Response:**

We greatly appreciate your helpful comment and suggestion.

There are some studies on the effects of the electric field and wind field disturbances during storms on ionospheric Es-layer and MLT region winds. However, it is the first

time to study the influence of storm on mesospheric metal atom layer through chemical processes, and there is not much helpful reference available.

According to your valuable comments and suggestions here and below, we have discussed and referenced existing atmospheric metal chemistry models more explicitly in the revision. The chemical processes of Na and Ni are added in the Discussion Section. We also find the Ca layer have experienced a greater depletion, and we have conducted the corresponding discussions. The correlation coefficients of the metal atom abundance variations are also a very interesting topic. But it is not the main topic of this manuscript, thus we have briefly analyzed why the correlation coefficient of Pingquan Na-Yanqing Ca is the lowest.

3. **Line 42-43:**

"It is widely accepted that chemical reactions below 85 km altitude have an important impact on the density variation of Na atoms, but the chemical reaction above 85 km has little influence."

This conflicts with later claims about chemical reactions influencing metal layers during storms at ~90-100 km. The authors should clarify how their findings contrast with or extend prior understanding.

**Response:**

Thanks very much for your kindly reminder and helpful comments.

The expression of this text can indeed lead to misunderstanding. We have revised this sentence and provided the correct and appropriate expression.

It is widely accepted that the Na layer exists at an altitude of 80-105 kilometers, and at lower altitudes, Na atoms are mostly converted into compounds due to chemical reactions.

4. **Line 55-58:**

Claiming that the storm cannot penetrate to metal layers is common but somewhat oversimplified. Some studies have shown ion-neutral coupling effects in MLT during storms — this could be acknowledged.

**Response:**

We appreciate your erudition and thank you for your kind suggestions.

We completely agree with you. We have revised this sentence and made the expression more rigorous.

Although the wind field and electric field disturbances are hard to penetrate deep into the dense atmospheric region due to the viscous force of air, during storms, the wind field disturbances in Mesosphere and Low Thermosphere (MLT) region as well as the plasma variations in ionospheric E-region have been recorded and analyzed (Chen et al., 2023; Li et al., 2019; Li et al., 2024; Resende et al., 2020).

Thanks again for your valuable suggestion.

**Reference**

Li, J., Wang, W., Lu, J., Yue, J., Burns, A. G., Yuan, T., et al. A modeling study of the responses of mesosphere and lower thermosphere winds to geomagnetic storms at middle latitudes. Journal of Geophysical Research: Space Physics, 124(5), 3666–3680. https://doi.org/10.1029/2019JA026533, 2019.

Resende, L. C. A., Shi, J. K., Denardini, C. M., Batista, I. S., Nogueira, P. A. B., Arras, C., et al. The influence of disturbance dynamo electric field in the formation of strong sporadic E layers over Boa Vista, a low-latitude station in the American sector. Journal of Geophysical Research: Space Physics, 125(7), e2019JA027519. https://doi.org/10.1029/2019JA027519, 2020.

5.  **Line 58-59:**

"Whether a storm can affect the atmospheric layers as low as the Mesosphere and Low Thermosphere (MLT) region is a very interesting topic."

This is true, but not a novel question. The authors should better situate their work in the context of previous MLT storm effects research.

**Response:**

We greatly appreciate your insights in this field.

The influence of storms on MLT region winds and ionospheric Es has been studied and reported. Our studies differ from the them. We have found the storm affected the metal atom layer in the MLT region through chemical reaction. The context of the manuscript has been revised to avoid misunderstanding.

6. **Line 64:**

Statement that this is the first time a storm effect on metal layer has been introduced is somewhat strong. The work is likely first strong evidence of depletion caused by changed chemistry, but literature on sporadic E-layer and metal ion responses exists. This should be framed carefully.

**Response:**

Thanks so much for your helpful comments.

We completely understand your comments. There have are several studies on the effects of magnetic storms on the ionospheric sporadic E-layer (Es), including our own article. And it is well known that ionospheric Es is composed by metal ions. Thus, we have revised this context accordingly.

"... and it is the first time to introduce the influence of storm on Earth's metal layer." is revised to "... and this is the first time the influence of a storm on the metal atoms in the MLT region is recorded.".

7. **Line 69-76:**

Description of lidar systems needs clarity on detection limits, uncertainty, and calibration methods. Are all metal densities quantitative and directly comparable?

**Response:**

Thank you very much for your helpful suggestion and question.

In the revision, we have added the information of the detection limits, uncertainty, and calibration methods of the lidars as following.

We define the detection threshold of the lidars as the background noise plus two times the noise standard deviation (Gao et al., 2015). The error of the lidar photon counting is inversely proportional to the square root of the return photon number, and the error of the measured metal atom density determined by the error of the photon counting is no more than 5%.

The absolute density can be estimated by the following standard lidar equation (Megie et al., 1978; Gerding et al., 2018; Chu and Papen, 2005),

$$n_{atom}(z) = n_R(z_R) \frac{N_s(\lambda, z) - N_B \Delta t}{N_R(\lambda, z_R) - N_B \Delta t} \frac{z^2}{z_R^2} \frac{\sigma_R(\lambda)}{\sigma_{atom}(\lambda)}$$

The reference altitude $z_R$ is usually selected between 30-45 km, and $z$ represents the altitude of the resonant fluorescence scattering region, and $n_R(z_R)$ is the atmospheric total number density at the reference altitude, which can be obtained from the NRLMSISE-00 model. $N_s(\lambda, z)$ and $N_R(\lambda, z_R)$ are the return photon numbers from the altitude $z$ and the reference altitude $z_R$, respectively. $N_B$ is the return photon number from the noise altitude generally at 150-190 km. $\lambda$ represents the wavelength of Na (589.158 nm), Ca (422.6728 nm), and Ni (341.5744 nm). At 30 km altitude, the Rayleigh backscattering cross section $\sigma_R(589.158nm)$ for Na is $4.14\times10^{-32}$ m$^2$sr$^{-1}$ and the effective backscattering cross section of resonance $\sigma_{atom}(589.158nm)$ for Na is $5.22\times10^{-16}$ m$^2$sr$^{-1}$, the $\sigma_R(422.6728nm)$ for Ca is $1.28\times10^{-31}$ is m$^2$sr$^{-1}$ and the $\sigma_{atom}(422.6728nm)$ for Ca is $2.23\times10^{-16}$ m$^2$sr$^{-1}$. At 40 km altitude, the $\sigma_R(341.5744nm)$ for Ni is $3.55\times10^{-31}$ m$^2$sr$^{-1}$ and the $\sigma_{atom}(341.5744nm)$ is $1.58\times10^{-17}$ m$^2$sr$^{-1}$. When calculating the absolute density of the Ni layer, it is also necessary to consider the branching fraction of the transition lines of the received return photons (Wu et al., 2021).

The number of return photons should be proportional to the flux of the transmitted laser photons. However, for large laser intensities, the returned fluorescence photons are no longer proportional to the laser power, but less than expectations. Thus, the saturation effect occurs. The content of Ni atoms in metal layer is relatively low, so its saturation effect is generally not considered. The energy of Na lidar's laser is not high enough to produce saturation effect. The laser energy emitted by the Ca lidar is relatively large and the divergence angle of the laser beam is very small, thus the saturation effect of the Ca lidar must be considered. Refer to the method of Megie et al., (1978), Welsh and Gardner (1989), and Chu and Papen (2005) to estimate saturations, the ratio $\frac{N^{Sat}(z)}{N(z)} = 91.42\%$ is obtained (Wu et al., 2020), where $N^{Sat}(z)$ is the number of the photons received at height $z$, when the saturation effect occurs, and $N(z)$ is the number of photons received at height $z$, when there is no saturation effect. Thus, we have taken this ratio into account in the effective scattering cross sections to correct the Ca densities.

All the measured metal densities, including the density values of different metal atoms at different places, are quantitative and can be compared directly.

In the revision, we have briefly introduced the information above. Thank you for your valuable suggestions again!

**Reference**

Chu, X. and Papen, G. C.: Resonance fluorescence lidar for measurements of the middle and upper atmosphere, in Laser Remote Sensing, Fujii, T. and Fukuchi, T., eds. (CRC Press), 179–432, 2005.

Gao, Q., Chu, X., Xue, X., Dou, X., Chen, T., and Chen, J., Lidar observations of thermospheric Na layers up to 170 km with a descending tidal phase at Lijiang (26.7°N, 100.0°E), China, J. Geophys. Res. Space Physics, 120, 9213-9220, doi:10.1002/2015JA021808, 2015.

Gerding, M., Daly, S., and Plane, J.M.C.. Lidar Soundings of the Mesospheric Nickel Layer Using Ni(3F) and Ni(3D) Transitions. Geophys. Res. Lett. 46. doi:10.1029/2018GL080701, 2018.

Megie, G., Bos, F., Blamont, J. and Chanin, M.: Simultaneous nighttime lidar measurements of atmospheric sodium and potassium, Planet. Space Sci. 26, 27-35, https://doi.org/10.1016/0032-0633(78)90034-X, 1978.

Picone, J. M., Hedin, A. E., Drob, D. P., and Aikin. A. C.: NRLMSISE-00 empirical model of the atmosphere: Statistical comparison and scientific issues. J. Geophys. Res. 107(A12): 1468. doi:10.1029/2002JA009430, 2002.

Welsh, B. and Gardner C.: Nonlinear resonant absorption effects on the design of resonance fluorescence lidars and laser guide stars, Appl. Opt., 28, 4141-4153, https://doi.org/10.1364/AO.28.004141, 1989.

Wu, F., Zheng, H., Cheng, X., Yang, Y., Li, F., Gong, S., Du, L., Wang, J., Yang, G., Simultaneous detection of the Ca and $Ca^+$ layers by a dual-wavelength tunable lidar system, Appl Opt., 59(13), 4122-4130, doi: 10.1364/AO.381699, 2020

Wu, F. J. et al.: Lidar observations of the upper atmospheric nickel layer at Beijing (40°N,116°E). Journal of Quantitative Spectroscopy and Radiative Transfer, 260, 107, 468. https://doi.org/10.1016/j.jqsrt.2020.107468, 2021.

8.  **Line 94-113 (Geomagnetic Indices):**

The storm description is adequate. However, consider adding direct linkage of timing of substorm peaks with metal layer depletion timing for clarity.

**Response:**

Very good suggestion.

We did not observe the process of the metal layer depletion. As the observations in the red box in Figure 2, the metal layers have been depleted in the night of 4 Nov.. The simulated oxygen density variations in Figure 6 show that the O density enhancement began at 3:00 UT on 4 November 2021. Thus, we speculate that the dissipation of the metal layer began in the daytime of the 4 Nov., but at that time the lidar could not work. We have added an up arrow at ~2:30 UT of 4 Nov. in Figure 1e to indicate the beginning of the simulated O density enhancement in MLT region.

[Figure]

**Figure 1. Geomagnetic indexes. (a) Bz, (b) SYM/H, (c) AE, (d) PC and (e) kp indexes on 3-5 November 2021. The green and red dashed boxes indicate the observation periods of the lidars in the nights of 3 and 4 Nov.. The red up arrow in the bottom plot is used to indicate the beginning of the simulated O density enhancement in MLT region.**

9. **Line 167-176:**

The statistical argument that simultaneous depletion at 4 stations with low chance of random occurrence ($p=1.334\times10^{-7}$) supports storm causality is good.

However, the low correlation coefficients between metal layers (e.g. $3.578\times10^{-4}$ for Pingquan-Na and Yanqing-Ca) invite discussion — why do metal species respond differently? Is variability noise or physical reason? This warrants explanation.

**Response:**

Very good question! We appreciate your carefulness and insightful comments.

Table 1. Cross-correlation coefficients of different data

| Item | Pingquan Na-Yanqing Na | Pingquan Na-Yanqing Ca | Pingquan Na-Yanqing Ni | Yanqing Na-Ca | Yanqing Na-Ni | Yanqing Ca-Ni |
|---|---|---|---|---|---|---|
| Correlation coefficient | 0.397 | $3.578\times10^{-4}$ | 0.507 | 0.346 | 0.603 | 0.169 |

All the correlation coefficients of the metal atom abundance variations are relatively low, indicating there are no much relationship between them. However, this does not mean that they are not correlated over a longer time scale, such as one year (Höffner and Friedman, 2004). We also can find that the abundance variation of the Ca has presented less correlation with other atoms, and the correlation coefficient of Pingquan Na-Yanqing Ca is only $3.578\times10^{-4}$. Observations indicate that the Ni and Na layers show close correlations on the scale of hours (Wu et al., 2022). And the calcium layer is somewhat unique. It find that although Ca abundance has a similar elemental abundance to Na in meteorites, the Ca atom abundance in metal layer is roughly 2 orders of magnitude smaller than Na. Plane et al., (2018) suggested that CaOH and $CaCO_3$ are stable reservoirs for Ca in metal layer, as a result, more Ca atoms are converted to compound and the Ca

atom abundance in metal layer is much less than the Na atom abundance. The Ca abundance variation is more affected by chemical reactions, so it has less correlation with other metal atoms. The relevance between the Ca and Na in different places further decrease, thus the estimated correlation coefficient is very low.

This is the our speculation based on the limited observations. It is not the focus of this manuscript and we do not intend to discuss more about this topic in this manuscript.

**Reference**

Höffner, J. and Friedman, J. S.: The mesospheric metal layer topside: a possible connection to meteoroids, Atmos. Chem. Phys., 4, 801-808, https://doi.org/10.5194/acp-4-801-2004, 2004.

Plane, J. M. C., Feng, W. H., Gómez Martín, J. C., Gerding, M., and Raizada, S.: A new model of meteoric calcium in the mesosphere and lower thermosphere. Atmospheric Chemistry and Physics, 18(20), 14,799–14,811. https://doi.org/10.5194/acp-18-14799-2018, 2018.

Wu, F., Chu, X., Du, L., Jiao, J., Zheng, H., Xun, Y., et al.: First simultaneous lidar observations of thermosphere-ionosphere sporadic Ni and Na (TISNi and TISNa) layers (~105-120 km) over Beijing (40.42°N, 116.02°E). Geophysical Research Letters, 49, e2022GL100397. https://doi.org/10.1029/2022GL100397, 2022.

10. **Line 209-250:**

The TIEGCM model description and simulated O/N2 enhancements support the scenario. It is a limitation that TIEGCM bottom boundary is ~97 km, slightly above some metal layers' lower height ~80 km. Authors should discuss how simulation resolution affects conclusions and chemical coupling.

**Response:**

Thanks so much for your insightful suggestions.

Yes, the bottom boundary of TIEGCM is ~97 km, solving a series of equations in the thermosphere and ionosphere system self-consistently. Thus, Figures 6a and 6c display the O variations at lower heights (~97-104 km). Figure 6a shows the enhancement of the $O/N_2$ ratio between 100-200 km altitude, which was due to the increase of O as well as the decrease of $N_2$. Additionally, Figure 6c shows that at the lower heights, the O density

increased even higher and earlier. The O increases in lower thermosphere were bound to influence the O variations in the metal layer at an altitude of 80-100 km by molecular diffusion and downwelling, resulting in the O density increase there.

11. **Line 252-291:**

The detailed chemical pathway for Ca compounds affecting Ca atom abundance is well presented (Equations 1-6).

For a stronger theoretical argument, extend these to Na, Ni chemistry or provide reasoning why Ca is emphasized.

**Response:**

Very good suggestion and comment.

Metallic elements are not stable in the Earth's atmosphere, and they will constantly shift between ions, atoms, and compounds. The released Na atoms from meteoroids will be oxidized by $O_3$ to form NaO (R1) in MLT region, and further react with $H_2O$ or $H_2$ to form NaOH (R2 & R3). $NaHCO_3$ is the recombination of NaOH with $CO_2$ (R4) and it is the major reservoir for Na on the bottom side of the metal layer (Gómez-Martín et al., 2017; Plane et al., 2015; Yuan et al., 2019). $NaHCO_3$ is converted back to Na either by photolysis ($hv$) or by reaction with atom H (R5 & R6). While the O density increases, more Na atoms will become $NaHCO_3$ to reduce the Na density.

$$Na + O_3 \rightarrow NaO + O_2 \tag{R1}$$

$$NaO + H_2O \rightarrow NaOH + OH \tag{R2}$$

$$NaO + H_2 \rightarrow NaOH + H \tag{R3}$$

$$NaOH + CO_2 \rightarrow NaHCO_3 \tag{R4}$$

$$NaHCO_3 + hv \rightarrow Na + HCO_3 \tag{R5}$$

$$NaHCO_3 + H \rightarrow Na + H_2CO_3 \tag{R6}$$

Ni atoms in MLT region is oxidized by $O_3$ and $O_2$ to form NiO and $NiO_2$, respectively (R7 and R8). These two Ni compounds further react with $O_3$, $O_2$, $CO_2$, and $H_2O$ to form higher oxides, carbonates, and hydroxides (e.g. R9-R14), therein, NiOH, Ni $(OH)_2$ and $NiCO_3$ is the major reservoir for Ni. These higher compounds will be converted to NiOH and NiO, which are finally converted back to Ni by reaction with atom H and O, as well as CO (R15-R17) (Daly et al., 2020). The increased O will transform more Ni atoms into the compound reservoir to dissipate the Ni layer.

$$Ni + O_3 \rightarrow NiO + O_2 \qquad (R7)$$

$$Ni + O_2 (+M) \rightarrow NiO_2 \qquad (R8)$$

$$NiO + O_3 \rightarrow NiO_2 + O_2 \qquad (R9)$$

$$NiO + O_2 \rightarrow ONiO_2 \qquad (R10)$$

$$NiO + CO_2 (+M) \rightarrow NiCO_3 \qquad (R11)$$

$$NiO + H_2O (+M) \rightarrow Ni(OH)_2 \qquad (R12)$$

$$Ni(OH)_2 + H \rightarrow NiOH + H_2O \qquad (R13)$$

$$NiO_2 + O_3 \rightarrow ONiO_2 + O_2 \qquad (R14)$$

$$NiOH + H \rightarrow Ni + H_2O \qquad (R15)$$

$$NiO + O \rightarrow Ni + O_2 \qquad (R16)$$

$$NiO + CO \rightarrow Ni + CO_2 \qquad (R17)$$

Na and Ni chemistry as presented above have been added in the revision. Thanks so much for your helpful suggestion.

The increased O atoms during the storm will also participate in the chemical cycle of the metal elements in metal layer and put more metal atoms in a compound state, thus lead to the depletion of the metal layer.

The metal atoms, including Na, Ca, Ni, Fe and etc, in metal layer will react with oxygen to form compounds, and some compounds will convert back into atoms. Therein, Ca is somewhat different from other metal elements. Although Ca has a similar elemental abundance to Na in meteorites, the Ca atom abundance in metal layer is roughly 2 orders of magnitude smaller than Na. Plane et al., (2018) suggested that CaOH and $CaCO_3$ are stable reservoirs for Ca, as a result, the increased O density converts more Ca atoms to compounds, and thus the Ca density in the metal layer decreases significantly as shown in Figure 2.

**Reference**

Daly, S. M., Feng, W., Mangan, T. P., Gerding, M., & Plane, J. M. C.: The Meteoric Ni Layer in the Upper Atmosphere. Journal of Geophysical Research: Space Physics, 125, e2020JA028083. https://doi.org/10.1029/2020JA028083, 2020.

Gómez Martín, J. C., Seaton, C., de Miranda, M. P., and Plane, J. M. C.: The Reaction between Sodium Hydroxide and Atomic Hydrogen in Atmospheric and Flame

Chemistry, The J. Phys. Chem. A, 121, 7667-7674, https://doi.org/10.1021/acs.jpca.7b07808, 2017.

Plane, J. M. C., Feng, W., and Dawkins, E. C. M.: The mesosphere and metals: Chemistry and changes, Chem. Rev., 115, 4497-4541, https://doi.org/10.1021/cr500501m, 2015.

Plane, J. M. C., Feng, W. H., Gómez Martín, J. C., Gerding, M., and Raizada, S.: A new model of meteoric calcium in the mesosphere and lower thermosphere. Atmospheric Chemistry and Physics, 18(20), 14,799–14,811. https://doi.org/10.5194/acp-18-14799-2018, 2018.

Yuan, T., Feng, W., Plane, J. M. C., and Marsh, D. R.: Photochemistry on the bottom side of the mesospheric Na layer, Atmos. Chem. Phys., 19, 3769-3777, https://doi.org/10.5194/acp-19-3769-2019, 2019.

12. **Line 293-312:**

Recommend highlighting limitations explicitly (measurement spatial coverage limited to mid-latitudes China, need for more latitude coverage).

Suggest stating potential implications on atmospheric chemistry and space weather forecasting more explicitly.

**Response:**

Thank you very much for your valuable suggestions that help us revise the conclusion section.

The second and third paragraphs of this section has been revised significantly according to your suggestion.

Magnetic storms affect the ionosphere and thermosphere by disturbing the electric field and wind field usually. Metal layers are located at a lower altitude, where the ambient atmospheric density is much greater. It is generally believed that the influence of magnetic storms is difficult to reach the altitudes of metal layer. Our observations present two very interesting phenomena, one is that the effects of magnetic storm have penetrated through the thermosphere and reached the MLT region, and the other is that the pathway of storm influence is no longer the usual electrodynamic process, but a chemical process. The storm has provided us a very nice opportunity to record the responses of the metal

layer to breaking of the chemical equilibrium in MLT region. It implies that magnetic storms have a broader impact on Earth's atmosphere, and a new horizon is opened up for us to study of the impact of magnetic storms on the Earth's space environment.

This observation was carried out only at mid-latitudes of China. However, what happens at other latitudes is unknown, especially more significant impact may occur at higher latitudes as Figure 4 indicated. Thus, it is necessary to record the storm influence on mesospheric metal layers at different latitudes for more latitude coverage. The new observation station in Mohe, Heilongjiang province, China (122.37°E, 53.5°N) is now in operation. We will report more observation results at higher latitude in the coming years.

**3. Grammatical, Syntax, and Stylistic Issues**

13. **Line 17:** "wind and electric field disturbances of a magnetic storm is hard" → "are hard" (plural subject-verb agreement)

**Response:**

Thank you very much for pointing out the error in subject-verb agreement.

The sentence on Line 17 is revised to "...the wind and electric field disturbances of a magnetic storm are hard to penetrate deep into the Earth's dense atmospheric region.".

14. **Line 20:** "metal layers were observed to deplete by three lidars" → "were observed to deplete by three lidars" is unclear. Better: "were observed to decrease by observations from three lidars"

**Response:**

Thanks so much for your helpful suggestion.

The sentence on Line 20 is revised to " ...the atmospheric metal layers were observed to decrease by observations from three lidars at the mid-latitudes of China.".

15. **Line 21:** "The Na, Ca and Ni densities on the storm day" → remove extra space after Na **Line 22:** "The O/N2 column density ratio ... was much higher than that on the quiet days," → "was much higher than on quiet days,"

**Response:**

We appreciate the your attentive reading.

The sentence on Line 21 is revised to "The Na, Ca and Ni densities on the storm day were significantly lower than those on other days in October and November.".

The extra space after Na has been removed.

The sentence on Line 22 is revised to "The O/N2 column density ratio observed by the Global Ultraviolet Imager (GUVI) on the storm day was much higher than on quiet days.".

16. **Line 35:** "turn into compounds, deposit on the Earth's" → "turn into compounds, depositing..." / or "turn into compounds that deposit..."
**Response:**

We appreciate the your keen eye for detail.

The sentence on Line 35 is revised to "Components of the metal layer turn into compounds that deposit on the Earth's surface in about four years...".

17. **Line 40:** "density of Na peaks in winter and reaches its lowest level in summer at all latitudes" → Add comma after "winter"
**Response:**

We thank you very much for this precise correction.

The sentence on Line 40 is revised to "The density of Na peaks in winter, and reaches its lowest level in summer at all latitudes...".

18. **Line 56:** "variations in ionosphere," → "variations in the ionosphere,"
**Response:**

We appreciate your careful reading and precise feedback.

The sentence on Line 56 is revised to "...variations in the ionosphere, and heating of the thermosphere.".

19. **Line 64:** "and it is the first time to introduce the influence of storm" → "and this is the first time the influence of a storm on the metal layer is introduced"
**Response:**

We are grateful to you for your valuable correction.

The sentence on Line 64 is revised to "...and this is the first time the influence of a storm on the metal layer is introduced.".

20. **Line 70:** "dual-wavelength simultaneous detection system is used for Na and Ni observation and the all-solid-state narrowband lidar is used to observe the Ca layer" → awkward; better split into two sentences

**Response:**

We are grateful to you for your helpful suggestion. This sentence has been split into two sentences as following.

The Na and Ni observations were implemented by the dual-wavelength simultaneous detection system (Du et al., 2020; Jiao et al., 2015; Wu et al., 2021). The all-solid-state narrowband lidar is used to observe the Ca layer (Du et al., 2023).

21. **Line 102:** "20:00 UT indicating beginning of the storm" → "indicating the beginning of the storm"

**Response:**

We would like to extend our gratitude for your meticulous reading of our manuscript.

The sentence on Line 102 is revised to "...at ~20:00 UT indicating the beginning of the storm and...".

22. **Line 124:** "metal atoms dataset can be accessed in" → "metal atoms dataset is accessible at"

**Response:**

We are grateful to you for raising this point regarding language precision.

The sentence on Line 124 is revised to "The metal atoms dataset is accessible at the Zenodo website (Xu and Chen, 2025).".

23. **Line 138-140:** Multiple phrasing issues with awkward expressions like "is lower than those on the reference days as well as those of month average" → "is lower than on reference days and monthly averages"

**Response:**

Thanks so much for your carefulness and helpful suggestion.

The sentence on Line 138 is revised to "The maximum density of the Ni layer in Fig. 2n was lower than 110 atoms·cm⁻³, also lower than on reference days in Figs. 2m and 2o, and much lower than monthly averages in Fig. 2p.".

The sentence on Line 147 is revised to "The maximum Na, Ca and Ni densities on the storm day in Figure 2q, 2s 2t were lower than on reference days and monthly averages.".

The sentence on Line 149 is revised to "The maximum density of the red Na density profile in Figure 2r is lower than the green and gray profiles.".

The sentence on Line 150 is revised to "The height distribution of red profile is narrower than the blue one, thus the Na layer density on the storm day was generally lower than on 3 Nov.".

The sentence on Line 225 is revised to " At 18:20 UT in Figure 5f, the $\Sigma O/N2$ began to decrease, but it is still higher than on the quiet days.".

The sentence on Line 294 is revised to" Though the density of atmospheric metal layers is varying day to day, the Na, Ca and Ni densities recorded by the Yanqing lidar on the storm day were significantly lower than on other days...".

24. **Line 169:** "but the lowest column abundances in all the four plots of Fig. 3 occurred on 4 Nov. 2021 as shown by the red bars." → "in all four plots of Fig. 3" (remove "the")

**Response:**

Thank you so much for pointing out this grammatical oversight.

The sentence on Line 169 is revised to"... but the lowest column abundances in all four plots of Fig. 3 occurred on 4 Nov. 2021 as shown by the red bars.".

25. **Line 200:** "vertical advection during the upwelling will reduce O and enhance $N_2$, thus lead to the depletion of the $O/N_2$." → "thus leads to the depletion"

**Response:**

We are grateful to you for your attentive review and for catching this grammatical error.

The sentence on Line 200 is revised to "Vertical advection during the upwelling will reduce O and enhance $N_2$, thus leads to the depletion of the $O/N_2$. .".

26. **Line 221:** "The simulated $\Sigma O/N_2$ over Yanqing and Pingquan is 1.0109 and the GUVI measurement is 1.0181..." → The values are very close; it would be better to say "is 1.0109, close to the GUVI measurement of 1.0181"

**Response:**

Thank you very much for your valuable input, which has helped us to improve the clarity and precision of the manuscript.

The sentence on Line 221 is revised to "The simulated $\Sigma O/N_2$ over Yanqing and Pingquan is 1.0109, close to the GUVI measurement of 1.0181...".

27. **Line 246:** "The largest increase was $1.9 \times 10^{10}/cm^3$ at 100 km altitude..." → write as "$1.9 \times 10^{10}$ cm$^{-3}$" with correct spacing and units

**Response:**

We sincerely thank you for your valuable comment on adhering to proper scientific notation. We also take the comments and suggestions of Reviewer #1 into consideration, and keep consistent with the units of the metal atoms.

The sentence on Line 247 is revised to "The largest increase was $1.9 \times 10^{10}$ atoms·cm$^{-3}$ at 100 km altitude at 14:00 UT."

The sentence on Line 133 is revised to "...more than 3600 particles/cm³." -> "...more than 3600 atoms·cm$^{-3}$.".

The sentence on Line 136 is revised to "...as 16.6 particles/cm³." -> "...as 16.6 atoms·cm$^{-3}$.".

The sentence on Line 137 is revised to "...exceeded 80 particles/cm³." -> "...exceeded 80 atoms·cm$^{-3}$.".

The sentence on Line 138 is revised to "...density exceeded 40 particles/cm³." -> "...density exceeded 40 atoms·cm$^{-3}$.".

The sentence on Line 139 is revised to "...lower than 110 particles/cm³." -> "...lower than 110 atoms·cm$^{-3}$.".

28. **Line 262:** "As a result, the increased O density converts more Ca atoms to compounds, and thus the Ca density in the metal layer decreases significantly." → merge last two clauses: "...to compounds, thus decreasing Ca density significantly."

**Response:**

We appreciate your helpful comment on improving the sentence flow.

The discussion section has been revised significantly. This sentence is revised as "As a result, the increased O density converts more Ca atoms to the stable compound reservoirs, thus the Ca layer has suffered much severer dissipation during the storm as shown in Figure 2. ".